# The current status and determinants of perceived ageism among community-dwelling older adults: A cross-sectional, multifactorial analysis

Jinlei Du[☯], Min Wang[☯], Xiaoling Wu, Yulian Wu, Ling Lei, Fang Cao, Chencong Nie[*],

Zigong Fourth People's Hospital, Sichuan Province, China

☯ These authors contributed equally to this work.
* suifengpiaoyao012@163.com

## Abstract

### Objective

To investigate the prevalence of perceived ageism among community-dwelling older adults and systematically analyze its determinants, thereby providing a reference for the management of mental health in this population.

### Methods

A cross-sectional survey was conducted from April 15–30, 2025, in 45 communities in Zigong City, Sichuan Province, China. A total of 500 questionnaires were distributed, and 484 valid responses were collected. The survey included a general information questionnaire and the Perceived Ageism Questionnaire (PAQ). LASSO (Least Absolute Shrinkage and Selection Operator) regression and multiple linear regression analyses were used to identify risk factors associated with perceived ageism.

### Results

The mean score on the Perceived Ageism Questionnaire was $16.68 \pm 3.70$. Multiple linear regression analysis revealed that educational level, vision status, and hearing function were significantly associated with perceived ageism among community-dwelling older adults.

### Conclusion

Perceived ageism among community-dwelling older adults is at a moderate level. Healthcare providers should develop targeted interventions based on these determinants to enhance mental health and well-being in this population.

**Data availability statement:** The data supporting the findings of this study are available in the Zenodo repository at https://zenodo.org/records/15948622.

**Funding:** The author(s) received no specific funding for this work.

**Competing interests:** The authors have declared that no competing interests exist.

## 1. Introduction

With the intensification of population aging, the role and status of older adults in society have undergone significant changes [1]. Older adults, as a special group, possess unique physiological, psychological, and social characteristics [2]. Due to declines in physical function, cognitive ability, and social interaction, they often face difficulties in social adaptation, which leads to a range of physiological and psychological problems, such as disability, dementia, depression, and even suicide, severely reducing their quality of life [3]. The community, as the primary living space for older adults, is not only the venue for their daily activities but also a crucial source of social interaction, daily care, and emotional support [4]. However, it is within this environment that ageism may also emerge, potentially affecting the quality of life of older adults.

Ageism, which targets older adults, involves stereotypes, prejudices, and discrimination based on age [5]. It encompasses both hostile and benevolent forms. Hostile ageism manifests as overt neglect or abuse, while benevolent ageism projects positive values such as wisdom and kindness onto older adults, potentially leading to overprotection and paternalism, which can undermine their autonomy [6]. This study aims to explore the prevalence of perceived ageism among community-dwelling older adults and analyze its determinants to provide a basis for targeted interventions to improve mental health and social adaptation in this population.

## 2. Methods

### 2.1. Study population

A cross-sectional survey was conducted from April 15–30, 2025, in 45 communities in Zigong City, Sichuan Province, China. The sample size was estimated using the formula $N = \left(\frac{Z\alpha_{/2}\,\sigma}{E}\right)^2$, where $Z_{a/2}$ was set at 1.96 (corresponding to a significance level of 0.05), $\sigma$ represented the population standard deviation, and E represented the width of the confidence interval. Based on preliminary survey results from patients meeting the inclusion and exclusion criteria, the estimated $\sigma$ was approximately 10. To ensure high precision in the research findings, the confidence interval width was set at ±1 unit. Substituting these values into the formula, it was determined that approximately 384 patients would need to be surveyed. Considering a 10% rate of invalid samples, the final sample size required was calculated to be 422 patients.

Inclusion criteria: Age ≥ 60 years; no intellectual disability and able to complete the questionnaire independently; voluntary participation with informed consent.

Exclusion criteria: History of psychological or psychiatric illness within the past three months; participation in other psychological research studies.

### 2.2. Ethics approval and consent to participate

This study was approved by the Ethics Committee of Zigong Fourth People's Hospital (approval number: EC-2023–073) and conforms to the National Statement on Ethical

Conduct in Human Research (2016). The study was conducted in accordance with the Declaration of Helsinki. All participants provided written informed consent to participate in the study. All interviews and surveys were conducted in private settings to ensure participant comfort and confidentiality. Participants were informed that their participation was entirely voluntary and that they could withdraw from the study at any time without any consequences. All collected information was processed anonymously to protect participant privacy.

## 2.3. Survey instruments

### 2.3.1. General information questionnaire.
The general information questionnaire was specifically developed for this study based on a comprehensive review of the literature and extensive clinical experience. It included demographic information (e.g., gender, age, educational level, marital status), health-related information (e.g., hearing function, vision status, number of comorbid chronic diseases), and socioeconomic information (e.g., monthly income, perceived community support, primary caregiver). For indicators such as self-rated aging attitude, family harmony, communication ability, there is currently no systematic specific assessment tool available. Therefore, in this study, a Numeric Rating Scale (NRS) was used to assess the above-mentioned symptoms among community-dwelling older adults. All older adults selected an appropriate number within the range of 0–10 based on their actual situation, with higher numbers indicating more severe symptoms or stronger abilities. A score of 0–3 indicates poor or mild, 4–6 indicates average, and 7–10 indicates severe or strong. The English version of the general information questionnaire is provided as supplementary material (Supplementary Material: General Information Questionnaire)

### 2.3.2. Perceived ageism questionnaire (PAQ).
The PAQ, developed by Brinkhof [7] in 2022, measures the degree of perceived ageism among older adults. The questionnaire consists of 8 items across two dimensions: negative perceived ageism (items 1, 2, 4, 6, 7) and positive perceived ageism (items 3, 5, 8). Items are scored on a 5-point Likert scale, ranging from 1 (never) to 5 (very often). The total score ranges from 8 to 40, with higher scores indicating higher levels of perceived ageism. The PAQ has demonstrated satisfactory psychometric properties, with a Cronbach's α coefficient of 0.63 for the total scale, 0.75 for the negative subscale, and 0.81 for the positive subscale.To date, no internationally recognized thresholds exist for categorising the severity of perceived ageism as measured by the PAQ. To describe the symptom level in our sample, we therefore converted the total score into a percentage of the maximum (40 points) and applied the following provisional classification: < 30% (<12 points) = mild, 30–60% (12–24 points) = moderate, and >60% (>24 points) = severe.

## 2.4. Data collection method

The research team provided detailed introductions to the study background, objectives, and methods to community residents' committees and healthcare service center staff. During the survey period, community-dwelling older adults independently completed and anonymously submitted the questionnaires, which were collected on-site. For individuals with severe physical disabilities or visual/auditory impairments, research staff assisted in completing the questionnaires through an interview format.

## 2.5. Quality control

Prior to the study, all research staff underwent systematic training in theoretical knowledge and practical skills, including standardized greetings for participants, strict adherence to inclusion/exclusion criteria, and uniform assessment standards for questionnaire completion. During the survey, leading questions or suggestive responses were avoided. Two researchers reviewed and screened all collected questionnaires, excluding those with incomplete data, multiple corrections, or unclear handwriting. The statistical analysis of survey data was reviewed by a statistician to ensure the scientific validity and accuracy of the results.

## 2.6. Statistical methods

Data from the questionnaires were double-entered into Epidata 3.0 by two researchers and analyzed using SPSS 25.0 and R. Descriptive statistics were used for categorical data (frequency) and continuous data (mean ± standard deviation).

LASSO regression was employed to select variables associated with perceived ageism. This method applies L1 regularization to shrink coefficients and select variables, effectively preventing multicollinearity and enhancing model generalizability. Variables with non-zero coefficients from LASSO were included in multiple linear regression analysis to assess their independent impact on perceived ageism. Before multiple linear regression, multicollinearity was assessed using variance inflation factors (VIF), with any variable having a VIF > 10 being excluded. Bootstrap resampling (1000 iterations) was used to assess the stability of regression coefficients, with narrow 95% confidence intervals (excluding zero) indicating stable estimates. All analyses were conducted at a significance level of $P < 0.05$.

## 3. Results

### 3.1. General Information of survey participants and LASSO-based variable selection for perceived ageism symptoms

In this survey, a total of 500 questionnaires were distributed, and 495 were collected. Among the 495 collected questionnaires, 5 had multiple and repeated corrections, 4 had illegible handwriting, and 2 had incomplete information. After discussion by the project team, these 11 questionnaires were excluded. Ultimately, 484 valid questionnaires were collected, with an effective response rate of 97%. To ensure the reliability of the study conclusions and the precision of the overall parameters, the project team decided to include all 484 valid questionnaires in the statistical analysis.

Among the 484 community-dwelling older adults, there were 223 individuals aged 60–69, 170 individuals aged 70–79, and 91 individuals aged ≥80. The sample included 275 males and 209 females. The mean score on the Perceived Ageism Questionnaire was 16.68 ± 3.70. The LASSO (Least Absolute Shrinkage and Selection Operator) analysis identified educational level, employment status, number of medications taken, primary caregiver, vision function, hearing function, source of health information, and self-rated communication ability as variables significantly associated with perceived ageism among community-dwelling older adults. For details, please refer to Table 1, Variables Selected by LASSO Analysis for Assessing Perceived Ageism Among Community-Dwelling Older Adults, and Fig 1, LASSO Coefficient Pathway for Variables Assessing Perceived Ageism Among Community-Dwelling Older Adults.The plot displays the coefficient trajectories of candidate predictors as the penalty parameter λ increases. At the optimal λ selected via 10-fold cross-validation, eight variables retained non-zero coefficients: educational level, employment status, number of medications, primary caregiver, vision function, hearing function, source of health information, and self-rated communication ability.

### 3.2. Multivariate analysis results: Impact of key predictors on perceived ageism based on LASSO-selected variables

Perceived ageism scores were used as the dependent variable, and the independent variables selected by LASSO analysis were included in the multiple linear regression model. Before conducting the multiple linear regression analysis, collinearity diagnostics were performed on the variables selected by LASSO. The tolerance of each model was > 0.1, and the variance inflation factor (VIF) was < 10, indicating no multicollinearity among the variables [8].

The results of the multiple linear regression analysis showed that educational level, hearing function, and vision status were significant predictors of perceived ageism among community-dwelling older adults. Detailed results are presented in Table 2 Multivariate Linear Regression Analysis of Perceived Ageism Among Community-Dwelling Older Adults.

**Table 1. Variables Selected by LASSO Analysis for Assessing Perceived Ageism Among Community-Dwelling Older Adults.**

| Variable | Group | Number | Perceived Age Discrimination Score | LASSO Coefficient | Included in Model |
|---|---|---|---|---|---|
| Age | 60-69<br>70-79<br>≥80 | 223<br>170<br>91 | 16.78±3.71<br>16.49±3.76<br>16.77±3.57 | 0.000 | No |
| Gender | Male<br>Female | 275<br>209 | 16.76±3.66<br>16.56±3.75 | 0.000 | No |
| Marital Status | Currently Married<br>Not Currrently Married | 399<br>85 | 16.54±3.70<br>17.32±3.65 | 0.000 | No |
| Educational Level | Primary school or below<br>Junior high school<br>Senior high school<br>College or above | 196<br>141<br>106<br>41 | 17.63±3.47<br>16.89±3.83<br>14.95±3.19<br>15.85±3.90 | −1.240 | Yes |
| Primary Caregiver | Spouse<br>Children<br>Other | 314<br>96<br>74 | 16.58±3.66<br>17.47±3.95<br>16.04±3.36 | 0.200 | Yes |
| Degree of Disability | Extreme functional impairment<br>Severe functional impairment<br>Moderate functional impairment<br>Minimal functional impairment<br>Physiological self-care | 14<br>37<br>59<br>300<br>74 | 16.07±3.73<br>17.38±4.03<br>16.53±3.28<br>16.52±3.77<br>17.19±3.53 | 0.000 | No |
| Smoking Status | Yes<br>No | 88<br>396 | 16.48±4.11<br>16.72±3.60 | 0.000 | No |
| Alcohol Consumption | Yes<br>No | 85<br>399 | 16.44±3.89<br>16.73±3.66 | 0.000 | No |
| Employment Status | Employed<br>Unemployed | 188<br>296 | 16.40±3.57<br>16.85±3.77 | −0.457 | Yes |
| Medical Payment Type | Urban-Rural Resident Basic Medical Insurance<br>Employee Medical Insurance<br>Other | 232<br>212<br>40 | 16.62±3.41<br>16.63±3.94<br>17.28±4.00 | 0.000 | No |
| Monthly Income | ≤3000 CNY<br>3000-5000 CNY<br>>5000 CNY | 217<br>159<br>108 | 16.70±3.50<br>16.50±3.73<br>16.90±4.04 | 0.000 | No |
| Primary Source of Income | Pension<br>Financial support from children<br>Other | 251<br>135<br>98 | 16.67±3.81<br>17.03±3.31<br>16.42±3.75 | 0.000 | No |
| Number of Children | ≤1<br>2-3<br>>3 | 164<br>259<br>61 | 16.73±4.17<br>16.53±3.48<br>17.16±3.20 | 0.000 | No |
| Primary Place of Residence | Rural<br>Urban | 190<br>294 | 16.76±3.48<br>16.62±3.84 | 0.000 | No |
| Frequency of Participation in Social Activities | Frequent<br>Occasional<br>Never | 229<br>166<br>89 | 16.62±3.59<br>16.43±3.98<br>17.26±3.39 | 0.000 | No |
| Number of Comorbid Chronic Diseases | ≤1<br>2-3<br>>3 | 308<br>169<br>7 | 16.52±3.61<br>16.96±3.88<br>16.71±3.14 | 0.000 | No |
| Number of Medications Taken | ≤1<br>2-3<br>>3 | 169<br>198<br>117 | 16.11±3.79<br>16.81±3.34<br>17.26±4.02 | 0.307 | Yes |
| Self-rated Family Harmony | Good<br>Average<br>Poor | 354<br>116<br>14 | 16.60±3.82<br>16.72±3.39<br>18.21±2.69 | 0.000 | No |

*(Continued)*

**Table 1.** (Continued)

| Variable | Group | Number | Perceived Age Discrimination Score | LASSO Coefficient | Included in Model |
|---|---|---|---|---|---|
| Self-rated Sleep Quality | Good<br>Average<br>Poor | 120<br>164<br>200 | 16.67±3.96<br>16.79±3.73<br>16.59±3.51 | 0.000 | No |
| Self-rated Physical Health | Good<br>Average<br>Poor | 228<br>204<br>52 | 17.12±3.16<br>16.28±3.82<br>16.27±5.00 | 0.000 | No |
| Frequency of Communication with Peers | Frequent<br>Occasional<br>Rare | 76<br>394<br>14 | 17.01±4.12<br>16.57±3.69<br>17.01±3.67 | 0.000 | No |
| Hearing Function | Normal<br>Impaired | 274<br>210 | 16.20±4.08<br>17.30±3.02 | −0.433 | Yes |
| Vision Status | Normal<br>Abnormal | 299<br>185 | 16.00±3.27<br>17.77±4.07 | −1.350 | Yes |
| Self-perceived Attitude Toward Aging | Positive<br>Neutral<br>Negative | 195<br>236<br>53 | 16.66±3.90<br>16.65±3.74<br>16.85±2.59 | 0.000 | No |
| Source of Health Information | Multimedia<br>Healthcare professionals<br>Other | 433<br>37<br>14 | 16.77±3.61<br>15.14±3.93<br>17.93±4.71 | −0.189 | Yes |
| Perceived Community Support | Sufficient<br>Average<br>Insufficient | 239<br>221<br>24 | 16.57±4.20<br>16.89±3.20<br>15.75±2.27 | 0.000 | No |
| Self-rated Communication Ability | Strong<br>Average<br>Poor | 6<br>391<br>87 | 17.17±3.60<br>16.51±3.88<br>17.37±2.65 | −0.014 | Yes |

## 4. Discussion

### 4.1. Perceived ageism among community-dwelling older adults are at a moderate level

Our study investigated the prevalence of perceived ageism among 484 community-dwelling older adults across 45 communities in Zigong City, with a mean perceived ageism score of 16.68±3.70. Given the lack of an internationally recognized standard for categorizing the severity of perceived ageism, we employed a percentage conversion method; with the maximum possible score on the Perceived Ageism Questionnaire being 40, the mean score of 16 translates to 40% (16/40), indicating that perceived ageism among this population are at a moderate level. This finding is lower than the 45.9% reported by Araujo et al [9]. in their study of older patients receiving treatment in a hospital setting. This discrepancy may stem from the distinct social interactions, psychological support, and role expectations present in community versus hospital environments.

Firstly, community settings offer older adults a richer array of social interactions and a broader social network. Within communities, older adults can actively participate in collective activities at senior centers, engage in daily conversations with neighbors, and maintain frequent contact with family members, thereby gaining emotional support and a sense of belonging [10]. This active social participation enhances their psychological resilience, enabling them to better cope with potential ageism and thus reducing their perception of ageism. In contrast, hospital environments are relatively closed-off, with older patients primarily interacting with healthcare providers and fellow patients, resulting in a more limited social scope [11]. Moreover, these patients often experience significant physical discomfort and psychological stress, which may heighten their sensitivity to ageism.

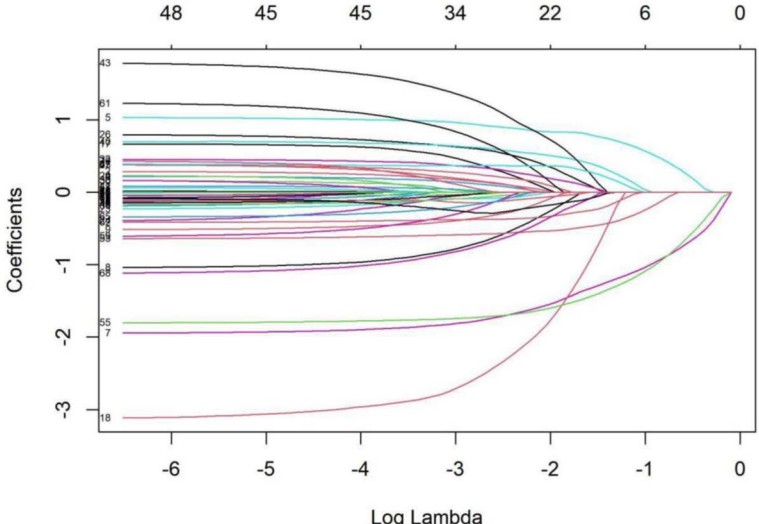

**Fig 1. LASSO Coefficient Pathway for Variables Assessing Perceived Ageism Among Community-Dwelling Older Adults.**

**Table 2. Multivariate Linear Regression Analysis of Perceived Ageism Among Community-Dwelling Older Adults.**

| Variable | B | SE B | T | P | 95%CI | Collinearity Diagnostics | |
|---|---|---|---|---|---|---|---|
| | | | | | | VIF | Tolerance |
| Educational Level | −0.936 | −0.248 | −5.414 | <0.001 | −1.275, -0.596 | 1.172 | 0.853 |
| Hearing Function | 0.800 | 0.107 | 2.266 | 0.024 | 0.106, 1.493 | 1.246 | 0.802 |
| Vision Status | 1.782 | 0.234 | 5.476 | <0.001 | 1.142, 2.421 | 1.019 | 0.981 |

Note: $F = 10.234, P < 0.001$, $R^2 = 0.238$, Adjusted $R^2 = 0.147$, Cohen's $f^2 = 0.312$.

Secondly, community environments are generally more inclusive of older adults. Community services and facilities are typically designed to better meet the actual needs of older individuals, such as accessible facilities and community medical points, which provide convenience and support, reducing feelings of neglect due to physical capabilities or social roles [12]. Conversely, the medical processes and operational rhythms in hospitals may not be as accommodating to older patients. For instance, complex registration systems, long waiting times, and the potential disregard for patients' opinions can exacerbate their sense of helplessness and discrimination [13].

Lastly, societal expectations of older adults differ between community and hospital settings. In communities, older adults are regarded as integral members of families and society, with their experience and wisdom respected. This positive role expectation helps to boost their self-esteem and sense of self-worth [14]. In hospitals, however, older adults are more often seen as individuals in need of care, which can lead to a sense of psychological marginalization.

Despite the lower prevalence of perceived ageism in community-dwelling older adults compared to those in hospital settings, the psychological impact of these symptoms should not be underestimated. Therefore, we recommend that healthcare providers and trained community workers conduct early screening and assessment of perceived ageism among community-dwelling older adults and establish a comprehensive and systematic clinical response framework to mitigate the psychological and physiological harm caused by perceived ageism and enhance the quality of life for older individuals.

## 4.2. Higher Educational Level Is Associated with Lower Perceived Ageism Risk Among Community-Dwelling Older Adults

Our findings indicate that older adults with higher educational attainment are less likely to perceive ageism in community settings. This association may be attributed to the impact of education on cognitive abilities, social participation, and psychological resilience. Firstly, higher education often correlates with stronger cognitive and information-processing skills. This enables older adults to more rationally assess instances of ageism in society. They can better comprehend the societal structures and cultural contexts that foster age-related stereotypes and use critical thinking to counteract these biases, thereby reducing their susceptibility to perceived ageism [15].

Secondly, individuals with higher education levels tend to have more extensive social participation and richer social networks. They are more likely to engage in community activities, volunteer services, and cultural learning, which not only enrich their lives but also enhance their sense of social identity and value [16]. Through active social interactions, they can gain more support and understanding, which in turn lowers their sensitivity to ageism.

Lastly, higher education contributes to greater psychological resilience [17]. It instills a stronger sense of self-efficacy and self-esteem, empowering older adults to more confidently express their needs and rights and reducing psychological stress and feelings of discrimination related to age [18]. Therefore, we should extend more care and assistance to older adults with lower educational levels. Proactive assessment of their psychological needs can facilitate early screening of perceived ageism and timely provision of necessary psychological support, as well as actively encouraging family involvement to enhance the sense of existence and respect among older adults, thereby reducing the risk of perceived ageism.

## 4.3. Vision and (or) Hearing Impairment Are Risk Factors for Perceived Ageism Among Community-Dwelling Older Adults

Our study revealed that older adults with vision and/or hearing impairments have a higher risk of experiencing perceived ageism. This heightened risk is likely due to the multifaceted impact of sensory impairments on social interactions, psychological states, and daily functioning. Firstly, vision and hearing impairments directly affect social interactions. Older adults with vision problems may struggle to recognize facial expressions and body language in social settings, while those with hearing difficulties may find it challenging to engage in conversations [19]. These communication barriers not only increase the likelihood of misunderstandings but also lead to feelings of social exclusion and neglect. For instance, older adults with hearing impairments may frequently interrupt others or misunderstand their intentions due to an inability to hear clearly, resulting in awkward situations and misunderstandings that they may attribute to ageism [20].

Secondly, sensory impairments can have a significant negative impact on psychological well-being. These impairments may lead to decreased self-confidence, feelings of helplessness and dependency, and increased self-doubt [21]. For example, older adults with vision impairments may feel frustrated by their inability to perform daily tasks independently, while those with hearing impairments may experience loneliness due to difficulties in socializing. This psychological vulnerability makes them more sensitive to perceived ageism, amplifying their perception of discrimination. Moreover, long-term sensory impairments can lead to anxiety and depressive symptoms, further reducing their psychological resilience and making them more susceptible to negative evaluations and discrimination from others.

Lastly, older adults with vision and/or hearing impairments face greater challenges in daily life. For example, vision impairments may require additional assistance for mobility, while hearing impairments may cause difficulties in understanding instructions. These daily inconveniences and dependencies may lead them to feel like a burden to others, thereby increasing their perception of discrimination [22]. For instance, an older adult with hearing impairments may feel confused when shopping if they cannot understand the cashier's instructions, and the cashier's impatience might be misinterpreted as ageism. These daily struggles can further exacerbate their feelings of being discriminated against.

In summary, the higher risk of perceived ageism among community-dwelling older adults with vision and/or hearing impairments is primarily due to the negative impacts of these sensory impairments on social interactions, psychological states, and daily functioning [23]. To address this issue, we recommend that communities provide targeted support services, such as vision and hearing rehabilitation training, guidance on the use of assistive devices, and early intervention by community-based psychological support groups. Additionally, public education should be strengthened to raise awareness and understanding of sensory impairments in older adults, reducing discrimination stemming from misunderstandings. Furthermore, family members and social workers should offer more support and understanding to help these older adults overcome daily challenges and improve their overall well-being and quality of life.

## 4.4. Implications for practice and policy

Based on our findings, we propose three mutually reinforcing areas for action. Community health services should integrate brief, routine screening for perceived ageism—particularly targeting older adults with lower education or sensory impairments—and embed clear referral pathways to psychological support. Local authorities can expand vision and hearing rehabilitation programmes, subsidise assistive devices, and train frontline community workers to recognise and counter ageist behaviours. Public health campaigns should leverage inter-generational activities and evidence-based educational materials to reshape societal narratives around ageing. Collectively, these measures can foster age-friendly environments that reduce perceived ageism and enhance older adults' mental well-being and social participation. Future research should adopt longitudinal designs to assess the long-term impact of such interventions and employ mixed-methods approaches to capture older adults' lived experiences of ageism in everyday contexts, thereby refining both policy and practice.In addition, to extend the current cross-sectional evidence, future research should adopt longitudinal designs to assess the long-term impact of such interventions and employ mixed-methods approaches to capture older adults' lived experiences of ageism in everyday contexts, thereby refining both policy and practice.

## 5. Conclusion

This study investigated the prevalence of perceived ageism among community-dwelling older adults in 45 communities in Zigong City. The results indicated that the symptoms of perceived ageism were at a moderate level and were significantly associated with educational level, vision status, and hearing function. However, the current clinical community has not paid sufficient attention to these symptoms, leaving many older adults vulnerable to mental health issues. Therefore, it is recommended that healthcare providers and trained community workers actively conduct early screening and assessment of perceived ageism among older adults to improve their mental health. Additionally, family members should be encouraged to participate in the psychological support of older adults, providing them with more love and a sense of belonging. Furthermore, we call on the general public to offer more tolerance and care to older adults, collectively creating an atmosphere conducive to the mental health of this population.

## 6. Study limitations

Despite our efforts to include a diverse sample, selection bias may have occurred in this study. The distribution of questionnaires across multiple communities may have unintentionally excluded or underestimated certain patient populations. This limitation suggests that the findings of our study may not be fully generalizable to all populations experiencing perceived ageism. To mitigate this, conducting a multicenter randomized sampling survey in future research could enhance the representativeness of the sample, address potential selection biases, and improve the generalizability of the findings.

Additionally, this study employed a cross-sectional design, which provides data at a single time point and does not allow for the assessment of temporal relationships between variables, making it difficult to establish causal links. Future research could adopt a longitudinal design to track patients over time and more accurately assess the causal relationships between risk factors and perceived ageism.

## Supporting information

**S1 Data. Raw data underlying the results presented in this study.**
(XLSX)

## Author contributions

**Conceptualization:** Jinlei Du, Min Wang.

**Data curation:** Jinlei Du, Min Wang, Yulian Wu, Ling Lei, Chencong Nie.

**Formal analysis:** Min Wang, Xiaoling Wu, Yulian Wu, Ling Lei, Fang Cao, Chencong Nie.

**Funding acquisition:** Jinlei Du, Min Wang, Xiaoling Wu, Fang Cao.

**Investigation:** Jinlei Du, Xiaoling Wu, Yulian Wu, Ling Lei, Fang Cao, Chencong Nie.

**Methodology:** Jinlei Du, Min Wang, Xiaoling Wu, Yulian Wu, Ling Lei, Chencong Nie.

**Project administration:** Jinlei Du, Yulian Wu, Fang Cao.

**Resources:** Jinlei Du, Min Wang, Xiaoling Wu, Chencong Nie.

**Software:** Jinlei Du, Min Wang, Xiaoling Wu, Yulian Wu, Ling Lei, Fang Cao, Chencong Nie.

**Supervision:** Yulian Wu, Ling Lei, Fang Cao.

**Validation:** Min Wang, Xiaoling Wu, Fang Cao, Chencong Nie.

**Visualization:** Jinlei Du, Min Wang, Ling Lei.

**Writing – original draft:** Jinlei Du, Min Wang, Xiaoling Wu, Chencong Nie.

**Writing – review & editing:** Jinlei Du, Chencong Nie.

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
