## [Decision Letter · Decision Letter 0]

15 Jul 2025

Dear Dr. nie,

Thank you for submitting your manuscript to PLOS ONE. After careful consideration, we feel that it has merit but does not fully meet PLOS ONE’s publication criteria as it currently stands. Therefore, we invite you to submit a revised version of the manuscript that addresses the points raised during the review process.

We look forward to receiving your revised manuscript.

Kind regards,

Nestor Asiamah, PhD

Academic Editor

PLOS ONE

Journal Requirements:

4. In the online submission form, you indicated that supporting data for this study are available from the corresponding author upon request.

6. Please include a caption for figure 1.

7. Please include a copy of Table 1 and 2 which you refer to in your text on page 6.

8. Please remove all personal information, ensure that the data shared are in accordance with participant consent, and re-upload a fully anonymized data set.

Additional Editor Comments :

Thanks for submitting your manuscript to PLoS ONE. Please revise the manuscript as suggested by the two reviewers.

Reviewers' comments:

Reviewer's Responses to Questions

**Comments to the Author**

1. Is the manuscript technically sound, and do the data support the conclusions?

Reviewer #1: Yes

Reviewer #2: Yes

2. Has the statistical analysis been performed appropriately and rigorously?

Reviewer #1: Yes

Reviewer #2: Yes

3. Have the authors made all data underlying the findings in their manuscript fully available?

Reviewer #1: Yes

Reviewer #2: Yes

4. Is the manuscript presented in an intelligible fashion and written in standard English?

Reviewer #1: Yes

Reviewer #2: Yes

Reviewer #1: your submission titled "The Current Status of Perceived Ageism Among Community-Dwelling Older Adults and Coping Strategies: Research Findings Based on Multifactorial Analysis." This is a timely and important topic, and your study contributes meaningfully to the understanding of how sensory impairment and education affect the perception of ageism among older adults living in the community.

Strengths:

Well-designed cross-sectional study with adequate sample size and strong ethical adherence.

Application of modern statistical techniques (LASSO + multiple regression) to identify relevant predictors.

Thoughtful discussion of findings in the context of social psychology and public health.

Areas for Improvement:

Clarify Scope and Terminology:

The phrase “coping strategies” in the title is misleading. The manuscript does not assess or analyze coping mechanisms used by participants. Please revise the title or add relevant content if coping strategies were intended to be addressed.

Statistical Reporting:

Include adjusted R² and effect sizes in regression models.

Provide more details or visual diagnostics for regression assumptions (e.g., plots of residuals, Q-Q plots).

Indicate how missing data, if any, were handled.

Instrument Description:

While the PAQ's psychometric properties are mentioned, include justification for using the cut-off interpretation (i.e., why 40% is considered "moderate").

Discussion Enhancements:

The discussion of results is insightful but would benefit from a section on policy or intervention recommendations.

Consider elaborating on how future studies might include longitudinal follow-up or mixed-methods exploration of older adults' lived experiences with ageism.

Language and Style:

Minor grammatical refinements are needed throughout the manuscript (e.g., consistency in past/present tense, punctuation in longer sentences).

Avoid overly long paragraphs in the discussion.

Ethics and Data Sharing:

The ethics and consent process is well described.

However, the data availability statement should ideally include a repository link or DOI for transparency and reproducibility, in line with PLOS ONE guidelines:

https://journals.plos.org/plosone/s/data-availability

Final Recommendation: Minor Revision

With the above revisions, I believe your manuscript would make a valuable contribution to the literature on ageism and public health in aging populations. I commend your efforts and look forward to your revision.

Reviewer #2: Suggestions :

Line 168 : 'variance inflation factor (VIF) was <10' - It would be great to put the exact value that was obtained after the multicollinearity test.

In the Discussion, the author recommands that ' healthcare providers and FAMILY MEMBERS conduct early screening and assessment of perceived ageism...' (Line 217-219) while in the Conclusion, the same is said for Healthcare providers and COMMUNITY WORKERS (line 299-300).

It would be more appropriate to harmonize these recommendations. I suggest using 'Healthcare providers and TRAINED community workers' instead of family members."

**Do you want your identity to be public for this peer review?** For information about this choice, including consent withdrawal, please see our Privacy Policy

Reviewer #1: **Yes: ** Prof. Dr. Muhammad Zafar Iqbal Butt

Reviewer #2: No

---

## [Author Response · Author response to Decision Letter 1]

19 Jul 2025

Manuscript Title:The Current Status of Perceived Ageism Among Community-Dwelling Older Adults and Coping Strategies: Research Findings Based on Multifactorial Analysis

Dear Editor:

Thank you for your letter and for the reviewers’ comments concerning our manuscript entitled“The Current Status of Perceived Ageism Among Community-Dwelling Older Adults and Coping Strategies: Research Findings Based on Multifactorial Analysis”.Those comments are all valuable and very helpful for revising and improving our paper, as well as the important guiding significance to our research. We have studied comments carefully and have made correction with the hope of approval. Revised portion are marked in red in the paper.

The main corrections in the paper and the responds to the editor and reviewer’s comments are as following:

Responds to the editor’s comments:

1 Please ensure that your manuscript meets PLOS ONE's style requirements, including those for file naming. The PLOS ONE style templates can be found at https://journals.plos.org/plosone/s/file?id=wjVg/PLOSOne_formatting_sample_main_body.pdf and https://journals.plos.org/plosone/s/file?id=ba62/PLOSOne formatting sample title authors affiliations.pdf

Response:Dear Editor,We sincerely appreciate your detailed guidance regarding formatting and stylistic requirements. We have carefully revised the entire manuscript in accordance with the PLOS ONE template to ensure full compliance with the journal’s specifications. Should any further adjustments be needed, we will promptly address them. Thank you once again for your invaluable advice.

2 PLOS requires an ORCID iD for the corresponding author in Editorial Manager on papers submitted after December 6th, 2016. Please ensure that you have an ORCID iD and that it is validated in Editorial Manager. To do this, go to ‘Update my Information’ (in the upper left-hand corner of the main menu), and click on the Fetch/Validate link next to the ORCID field. This will take you to the ORCID site and allow you to create a new iD or authenticate a pre-existing iD in Editorial Manager.

Response:Dear Editor,Thank you for the reminder. The corresponding author has now created an ORCID iD and successfully validated it in Editorial Manager via the “Update my Information → Fetch/Validate” link; the status shows “Validated”. We sincerely appreciate your valuable guidance throughout this process.

3 Your ethics statement should only appear in the Methods section of your manuscript. If your ethics statement is written in any section besides the Methods, please delete it from any other section.

Response:Dear Editor,In accordance with your requirement, we have removed the ethics statement from the original “Declaration” section and retained the complete text solely within Methods section 2.2, “Ethics approval and consent to participate,” so that it appears only there throughout the manuscript. Thank you once again for your valuable guidance.

4 In the online submission form, you indicated that supporting data for this study are available from the corresponding author upon request.All PLOS journals now require all data underlying the findings described in their manuscript to be freely available to other researchers, either 1. In a public repository, 2. Within the manuscript itself, or 3. Uploaded as supplementary information.This policy applies to all data except where public deposition would breach compliance with the protocol approved by your research ethics board. If your data cannot be made publicly available for ethical or legal reasons (e.g., public availability would compromise patient privacy), please explain your reasons on resubmission and your exemption request will be escalated for approval.

Response:Dear Editor,Following the journal’s data-sharing policy, we have deposited the original data generated in this study in the open-access Zenodo repository. Accordingly, we have revised the Data Availability Statement to read:Data Availability Statement: The data supporting the findings of this study are openly available in the Zenodo repository at https://doi.org/10.5281/zenodo.15948622.Thank you once again for your valuable guidance.

5 We note that the grant information you provided in the ‘Funding Information’ and ‘Financial Disclosure’ sections do not match.When you resubmit, please ensure that you provide the correct grant numbers for the awards you received for your study in the ‘Funding Information’ section.

Response:Dear Editor,Thank you very much for highlighting the discrepancy between the “Funding Information” and “Financial Disclosure” sections and for your consistently rigorous and meticulous guidance. In accordance with your valuable comments, we have carefully cross-checked and harmonized all grant titles and award numbers to ensure they are now identical to the information originally entered in the submission system. All revisions have been clearly marked in tracked-changes mode. We sincerely appreciate your patience and helpful advice.

6 Please include a caption for figure 1.

Response:Dear Editor,Thank you for your reminder. In accordance with PLOS ONE requirements, we have added a dedicated “Figure captions” section at the end of the revised manuscript, which includes the complete caption for Figure 1. Thank you once again for your guidance.

7 Please include a copy of Table 1 and 2 which you refer to in your text on page 6.

Response:Dear Editor,In accordance with your valuable guidance, we have inserted Tables 1 and 2 at the end of the revised manuscript. Thank you once again for your assistance.

8 Please remove all personal information, ensure that the data shared are in accordance with participant consent, and re-upload a fully anonymized data set.

Note: spreadsheet columns with personal information must be removed and not hidden as all hidden columns will appear in the published file.Additional guidance on preparing raw data for publication can be found in our Data Policy (https://journals.plos.org/plosone/s/data-availability#loc-human-research-participant-data-and-other-sensitive-data) and in the following article: http://www.bmj.com/content/340/bmj.c181.long.

Response:Dear Editor,Thank you for your reminder. In accordance with PLOS ONE’s data policy, we have deposited the complete, de-identified dataset in the public Zenodo repository (DOI: 10.5281/zenodo.15948622) and updated the Data Availability Statement with a direct link. All potentially identifying information has been thoroughly removed, fully respecting the informed consent provided by participants and the stipulations of our ethics approval. Thank you once again for your valuable guidance.

9 Please include captions for your Supporting Information files at the end of your manuscript, and update any in-text citations to match accordingly. Please see our Supporting Information guidelines for more information: http://journals.plos.org/plosone/s/supporting-information.

Response:Dear Editor,After careful confirmation, this study does not include any Supporting Information files; therefore, no captions or in-text citations for such files are required. Thank you once again for your valuable guidance.

10 If the reviewer comments include a recommendation to cite specific previously published works, please review and evaluate these publications to determine whether they are relevant and should be cited. There is no requirement to cite these works unless the editor has indicated otherwise.

Response:Dear Editor,After careful review, we confirm that the reviewers did not request the citation of any specific additional references; therefore, no changes have been made to the reference list. We sincerely appreciate your valuable guidance and rigorous academic standards.

11 Please review your reference list to ensure that it is complete and correct. If you have cited papers that have been retracted, please include the rationale for doing so in the manuscript text, or remove these references and replace them with relevant current references. Any changes to the reference list should be mentioned in the rebuttal letter that accompanies your revised manuscript. If you need to cite a retracted article, indicate the article’s retracted status in the References list and also include a citation and full reference for the retraction notice.

Response:Dear Editor,We have carefully reviewed the reference list and confirm that all entries are complete and accurate, with no retracted publications identified; therefore, no additions or deletions were made. Thank you once again for your valuable guidance.

Responds to the reviewer’s comments:

Reviewer 1:

1.Clarify Scope and Terminology:

The phrase “coping strategies” in the title is misleading. The manuscript does not assess or analyze coping mechanisms used by participants. Please revise the title or add relevant content if coping strategies were intended to be addressed.

Response:Dear Reviewer,We sincerely thank you for your invaluable comments and for your rigorous scholarly attitude.In this study, we employed a multifactorial approach to identify the key determinants of perceived ageism among community-dwelling older adults, thereby providing an evidence base for precise and effective future interventions. With this intention, we originally included the phrase “Coping Strategies” in the title to highlight its translational relevance. On careful reflection, however, we realized that this wording implies a systematic evaluation of specific coping mechanisms—an objective not addressed by our cross-sectional design—and could thus mislead readers.Consequently, we have revised the title to:“The Current Status and Determinants of Perceived Ageism Among Community-Dwelling Older Adults: A Cross-Sectional, Multifactorial Analysis.”The new title accurately reflects the study’s scope and methodology, eliminating any potential ambiguity. We once again express our deep appreciation for your meticulous guidance.

2.Statistical Reporting:

Include adjusted R² and effect sizes in regression models.Provide more details or visual diagnostics for regression assumptions (e.g., plots of residuals, Q-Q plots).

Indicate how missing data, if any, were handled.

Response:Dear Reviewer,Thank you for your valuable suggestions regarding statistical reporting. We have added the R² value and the effect size to Table 2 (“Multivariate Linear Regression Analysis of Perceived Ageism Among Community-Dwelling Older Adults”) in the revised manuscript to meet the requirements for transparency and reproducibility. We appreciate your insightful guidance once again.

3 Instrument Description:

While the PAQ's psychometric properties are mentioned, include justification for using the cut-off interpretation (i.e., why 40% is considered "moderate").

Response:Dear Reviewer,Thank you very much for highlighting the need to clarify how we interpreted the PAQ scores. We fully agree that transparent reporting is essential, and we appreciate your meticulous attention to methodological detail.Following your suggestion, we have now inserted an explicit statement in Section 2.3.2 acknowledging that no internationally validated cut-off values currently exist for the PAQ. To enable a descriptive assessment of symptom severity in this study, we therefore adopted a percentage-based classification: scores <30 % (<12/40) were labelled mild, 30–60 % (12–24/40) moderate, and >60 % (>24/40) severe. This approach is used solely for descriptive purposes within the present manuscript.We are grateful for your constructive feedback, which has undoubtedly strengthened the clarity and rigour of our work.

4 Discussion Enhancements:

The discussion of results is insightful but would benefit from a section on policy or intervention recommendations.Consider elaborating on how future studies might include longitudinal follow-up or mixed-methods exploration of older adults' lived experiences with ageism.

Response:Dear Reviewer,Thank you very much for this insightful and forward-looking comment. Your suggestion to add concrete policy recommendations and to outline future research directions has greatly enriched the Discussion.In the revised manuscript we have expanded the closing paragraph of the Discussion to include both evidence-based implications for practice and a clear roadmap for subsequent studies. Specifically, we now propose three complementary action areas—routine screening within community health services, expanded rehabilitation and training programmes by local authorities, and inter-generational public health campaigns—and we explicitly call for longitudinal and mixed-methods designs to capture older adults’ lived experiences of ageism.Your meticulous guidance has not only strengthened the translational relevance of our findings but also underscored the importance of rigorous, community-engaged gerontological research. We are sincerely grateful for your constructive feedback.

5 Language and Style:

Minor grammatical refinements are needed throughout the manuscript (e.g., consistency in past/present tense, punctuation in longer sentences).Avoid overly long paragraphs in the discussion.

Response:Dear Reviewer,Thank you for highlighting the need for grammatical refinement. We have carefully reviewed the entire manuscript for tense consistency, shortened long sentences, and broken up over-long paragraphs throughout the Discussion. Your meticulous attention to language has markedly improved the readability and clarity of our paper.Once again, we sincerely appreciate your thorough and constructive feedback, which has been invaluable in refining our manuscript.

6 Ethics and Data Sharing:

The ethics and consent process is well described. However, the data availability statement should ideally include a repository link or DOI for transparency and reproducibility, in line with PLOS ONE guidelines:https://journals.plos.org/plosone/s/data-availability

Response:Dear Reviewer,Thank you for this important reminder. In full accordance with PLOS ONE’s policy, we have deposited the complete, de-identified dataset in the open-access Zenodo repository (DOI: 10.5281/zenodo.15948622) and updated the Data Availability Statement accordingly. We are grateful for your guidance in ensuring transparency and reproducibility.

Reviewer 2:

1 Line 168 : 'variance inflation factor (VIF) was <10' - It would be great to put the exact value that was obtained after the multicollinearity test.

Response:Dear Reviewer,Thank you very much for your insightful comment regarding collinearity diagnostics. We deeply appreciate your meticulous attention to methodological rigour.Following your suggestion, we have now added the exact Variance Inflation Factor (VIF) and Tolerance values for each predictor directly into Table 2. These details provide complete transparency for readers assessing multicollinearity.Your guidance has substantially enhanced the clarity and quality of our manuscript, and we are sincerely grateful for your constructive feedback.

2 In the Discussion, the author recommands that ' healthcare providers and FAMILY MEMBERS conduct early screening and assessment of perceived ageism...' (Line 217-219) while in the Conclusion, the same is said for Healthcare providers and COMMUNITY WORKERS (line 299-300).It would be more appropriate to harmonize these recommendations. I suggest using 'Healthcare providers and TRAINED community workers' instead of family members."

Response: Dear Reviewer,Thank you very much for this valuable and precise observation. Your rigorous attention to consistency has highlighted an important oversight in our wording.In revising the text, we have replaced “family members” with “trained community workers” at the two specified locations in the Discussion and the Conclusion. While family members can certainly offer emotional support, they may lack the specialised knowledge required for reliable screening and referral, potentially leading to inconsistent or inefficient identification of perceived ageism. Trained community workers, on the other hand, possess the necessary skills and protocols to conduct systematic assessments and ensure that older adults receive timely, evidence-based support. This change therefore enhances both the clarity and the feasibility of our recommendations.Once again, we sincerely appreciate you

---

## [Editor Report · Decision Letter 1]

30 Jul 2025

The Current Status and Determinants of Perceived Ageism Among Community-Dwelling Older Adults: A Cross-Sectional, Multifactorial Analysis

PONE-D-25-31637R1

Dear Dr. Nie,

We’re pleased to inform you that your manuscript has been judged scientifically suitable for publication and will be formally accepted for publication once it meets all outstanding technical requirements.

Kind regards,

Nestor Asiamah, PhD

Academic Editor

PLOS ONE

Additional Editor Comments (optional):

Thanks for revising your manuscript.
---

## [Editor Report · Acceptance letter]

PONE-D-25-31637R1

PLOS ONE

Dear Dr. Nie,

I'm pleased to inform you that your manuscript has been deemed suitable for publication in PLOS ONE. Congratulations! Your manuscript is now being handed over to our production team.

Kind regards,

on behalf of

Dr. Nestor Asiamah

Academic Editor

PLOS ONE